# Warming underpins community turnover in temperate freshwater and terrestrial communities

Imran Khaliq [1,2,3,4] ✉, Christian Rixen [2,3], Florian Zellweger [5], Catherine H. Graham [5], Martin M. Gossner [5,6], Ian R. McFadden [5,6,7,8], Laura Antão [9], Jakob Brodersen [10,11], Shyamolina Ghosh [2,5,10,12], Francesco Pomati [1], Ole Seehausen [10,13], Tobias Roth [11,14], Thomas Sattler [15], Sarah R. Supp [16], Maria Riaz [17,18], Niklaus E. Zimmermann [5,6], Blake Matthews [10] & Anita Narwani [1] ✉

Rising temperatures are leading to increased prevalence of warm-affinity species in ecosystems, known as thermophilisation. However, factors influencing variation in thermophilisation rates among taxa and ecosystems, particularly freshwater communities with high diversity and high population decline, remain unclear. We analysed compositional change over time in 7123 freshwater and 6201 terrestrial, mostly temperate communities from multiple taxonomic groups. Overall, temperature change was positively linked to thermophilisation in both realms. Extirpated species had lower thermal affinities in terrestrial communities but higher affinities in freshwater communities compared to those persisting over time. Temperature change's impact on thermophilisation varied with community body size, thermal niche breadth, species richness and baseline temperature; these interactive effects were idiosyncratic in the direction and magnitude of their impacts on thermophilisation, both across realms and taxonomic groups. While our findings emphasise the challenges in predicting the consequences of temperature change across communities, conservation strategies should consider these variable responses when attempting to mitigate climate-induced biodiversity loss.

In response to rising temperatures, species are moving towards higher latitudes and elevations, or greater water depths to track their thermal requirements[1,2]. Such movements result in an increase in the prevalence of warm-affinity species in local communities over time, a phenomenon known as 'thermophilisation'[3–5]. The pace of thermophilisation is usually calculated as the temporal rate of change in the average thermal affinities of the species in the community (i.e., the community temperature index CTI)[6]. While thermophilisation is common in terrestrial and marine taxa, rates are variable among communities[3,6–13]. Moreover, systematic comparisons of community-level responses among realms are deficient, especially for the

freshwater realm, which is of particular interest due to its disproportionately high biodiversity per habitat area[14] (see refs. 9,12,15,16 for a few examples). By including a broader range of taxonomic groups and implementing comparative analyses between freshwater and terrestrial realms, we can identify possible reasons for variation in community responses to warming[17]. Systematic analyses of the drivers of thermophilisation are urgently needed to better develop effective conservation and management strategies.

The effects of warming vary both geographically and among taxa[1,2,8,13,18,19], suggesting that species- and community-level characteristics may influence the pace of thermophilisation. Therefore, we

expect that rates of thermophilisation may differ between the freshwater and terrestrial realms[17], and several biotic and abiotic predictors may lead to variation in rates of thermophilisation. First, large species will have stronger responses to rising temperatures compared to small species due to their greater metabolic constraints, and higher water and energy requirements for thermoregulation[20–22]. Moreover, the correlation between body size and temperature exhibits a more pronounced negative trend in aquatic ectotherms compared to their terrestrial counterparts, indicating notable distinctions between these two ecological realms[23]. Consequently, we anticipate a stronger positive association between mean body size of species in aquatic communities and temperature compared to the relationship in terrestrial communities, resulting in faster rates of thermophilisation in the former[24]. Second, we expect that the prevalence of thermal generalists in a community (i.e., species with large thermal niche breadths) should decrease thermophilisation[24], as they likely tolerate rising temperatures better than thermal specialists[25,26]. Further, body size and thermal niche breadth are often positively correlated[27] and may have an interactive influence on thermophilisation. Third, community species richness may reduce rates of thermophilisation[28] as diverse assemblages are likely to have greater resistance in the face of environmental change due to a greater likelihood of assemblages containing well-adapted species, greater biotic resistance to incoming species, and a lower influence of individual species on mean community properties[28–30]. Fourth, based upon previous reports, communities already experiencing warmer climates have shown slower rates of thermophilisation[13,26]. Thus, we expect communities with warmer baseline temperatures will have slower thermophilisation rates[7,31]. Finally, both immigration of species with high thermal affinities and extirpation of species with lower thermal affinities can lead to thermophilisation[32]. Hence, we investigated the extent to which these local gains and losses of species underpin observed rates of thermophilisation.

To quantify rates of thermophilisation, we compiled 13,324 time-series from several sources: Biodiversity Monitoring Switzerland (BDM, https://biodiversitymonitoring.ch), BioTIME[33], The United States Environmental Protection Agency (EPA)'s National Lakes Assessment (https://nationallakesassessment.epa.gov/), RivFishTIME[34] and several Swiss terrestrial insect studies[35–37]. In total, we included 158 distinct studies in addition to data belonging to BDM and EPA lakes. A total of 6201 terrestrial communities including plants (studies = 59 + BDM), birds (studies = 19 + BDM), insects (studies = 11 + BDM), and mammals (studies = 9), and freshwater communities ($n$ = 7123) of phytoplankton (studies = 2 + EPA), zooplankton (studies = 7 + EPA), insects (studies = 1 + BDM), and fish (studies = 50, Supplementary Fig. S1a, b). The communities comprised a total of 17,431 species, ranging from 4 to 666 species per community and spanning 5–38 years in the period 1980–2019. We calculated CTI values for each community as the mean of species' thermal affinities (species temperature index[3,5], "STI" hereafter) based on species' presence in the community. We then calculated thermophilisation rates as changes in CTI over time and related thermophilisation rates to changes in mean annual temperature. Additionally, we estimated the effects of (1) community average body size, (2) mean thermal niche breadth (calculated as the mean across species of the difference between the maximum and minimum temperatures experienced by each species across its range), (3) species richness and (4) baseline temperature, on the rates of thermophilisation. We included interactive effects of temperature change with all the other predictors. Finally, we investigated the relative contributions of local immigration (additions) and extirpation (losses) of species to rates of thermophilisation. In our findings, we highlight temperature change as the primary driver for thermophilisation rates in both freshwater and terrestrial ecosystems. Moreover, the interplay of body size, thermal niche breadth, species richness, and baseline temperature exhibited unique interactions with temperature, shaping the pace of thermophilisation across realms and taxonomic groups.

## Results and discussion

### Thermophilisation rates

Around half (52.99%) of the communities showed positive thermophilisation rates indicating shift in communities' composition towards prevalence of warm-affinity species. The observed rates are within the range of reported rates of thermophilisation for different taxa[8–13,38]. We found no difference in the rates of thermophilisation between realms (β = −0.009, $t$ value = −0.8, $p$ = 0.41, Fig. 1a), or between the different taxonomic groups within realm (Fig. 1b, Freshwater: $F$ = 2.4, $p$ = 0.09, Terrestrial: F = 0.46, $p$ = 0.62). Five out of six taxonomic groups had positive rates of thermophilisation, namely plants, birds, and terrestrial insects in the terrestrial realm (Fig. 1b) and aquatic insects and fish in the freshwater realm (Fig. 1b), similar to previous reports[9,12,15,16]. Zooplankton and phytoplankton communities were a clear exception, displaying cryophilisation, i.e., increased prevalence of cold-affinity species, and with negative rates an order of magnitude greater compared to other taxonomic groups (Fig. 1b, Supplementary Fig. S2). However, the results for phytoplankton and mammals need to be interpreted with caution because of a high proportion of phytoplankton species with a low number of occurrence records (for the estimation of STIs) and the low number of communities for mammals. Due to these reasons, we moved results of phytoplankton and mammals to the supplementary files.

### Predictors of thermophilisation at realm level

Predictors had various effects on the direction and magnitude of thermophilisation in the two realms. As expected, temperature change had an overall positive effect on thermophilisation rates in both realms but its effect was stronger for terrestrial communities (Fig. 2a, e; Supplementary Fig. S3, Supplementary Tables S1–S2). This may be partly explained by the weaker temperature change observed at the freshwater sites within our time-series (Supplementary Fig. S1c). For instance, if the change in temperature remains below the optimum for the majority of species it would not prompt organisms to respond and relocate to new areas, and could thus lead to a comparatively lower overall response. The impact of temperature change on thermophilisation rates differed between the two realms mainly via a three-way interaction involving thermal niche breadth and body size (Fig. 2a, e). In freshwater environments, communities with both broader mean thermal niches and larger mean body sizes responded more slowly to rising temperatures, while terrestrial communities with these attributes had faster responses to temperature change (Fig. 2a, e). This divergence between realms suggests a better ability of freshwater communities with broader thermal niches and larger body sizes to tolerate temperature increases, and maintain similar composition. In contrast, the same traits led to quicker community turnover on land, potentially indicating that metabolic or mobility constraints due to large body size play a more significant role in responding to temperature change in terrestrial systems[39]. This is further reinforced by the contrasting direction of this interactive effect on thermophilisation rates, which was negative in the freshwater realm, but positive in the terrestrial realm (Fig. 2a, e). Furthermore, on land the impact of temperature change on thermophilisation rates was influenced by baseline temperature, thermal niche breadth and species richness, with warmer baseline conditions, broader thermal niche breadth and greater species richness mitigating the effect of temperature change (i.e., underpinning slower rates of thermophilisation; Fig. 2a). The dampening effect of species richness on thermophilisation supports the hypothesis that diverse communities are more stable in response to environmental change[28,29], also confirming the potential insurance value of biodiversity in the provisioning of ecosystem functions and services[30]. The stabilising effects of species richness

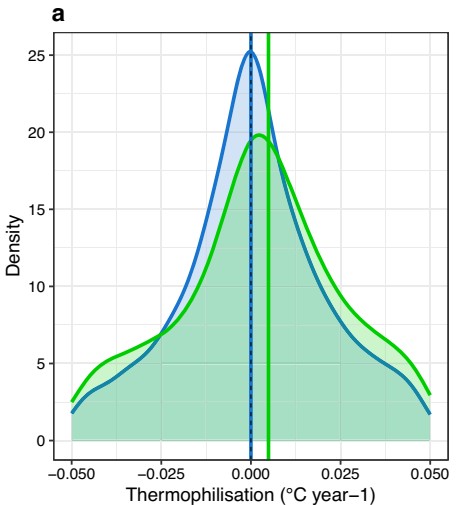

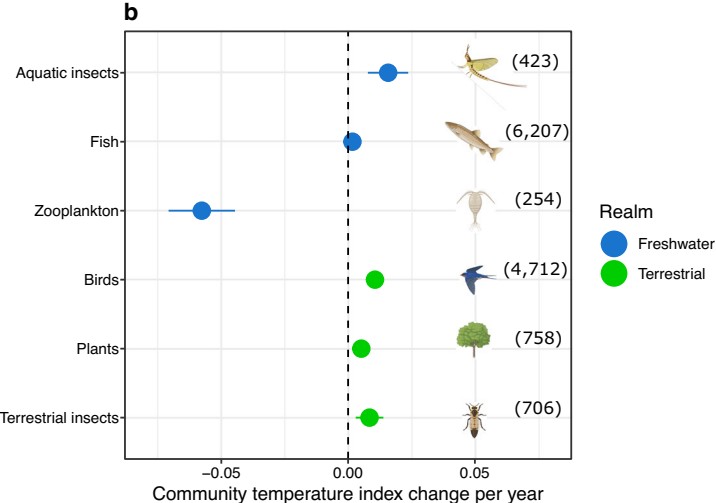

**Fig. 1 | Thermophilisation rates across realms and taxonomic groups. a** Density plots show the distribution of rates of thermophilisation in the freshwater and the terrestrial realms. Blue and green vertical lines represent the median values per realm. **b** Mean thermophilisation rates by taxonomic group, showing the change in the community temperature index (CTI) over time, estimated for each community and averaged across communities within each taxonomic group (numbers in parentheses indicate the number of time-series for each taxonomic group). Rates are significantly different from zero for all groups. Error bars represent the 95% confidence intervals of the mean. Silhouettes were created with BioRender.com.

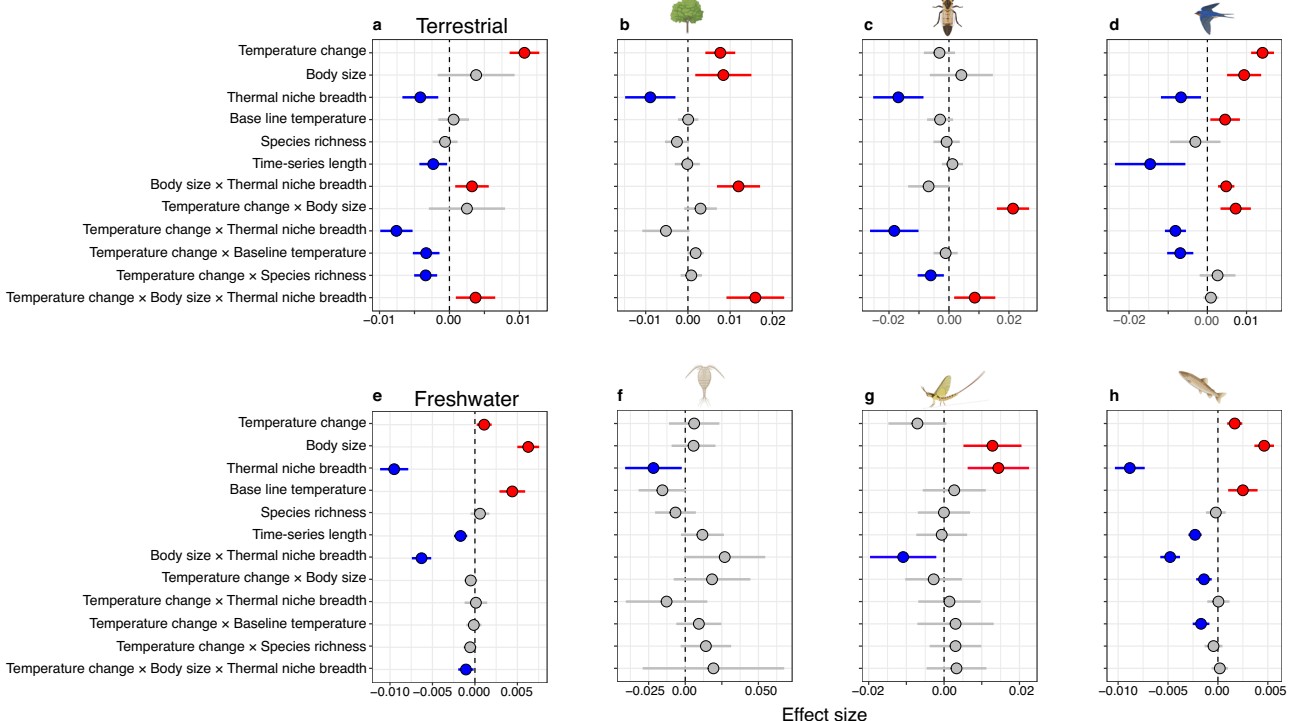

**Fig. 2 | Predictors of thermophilisation rates across realms and taxonomic groups.** The points show mean effect sizes of temperature change, average community body size, mean thermal niche breadth, baseline mean annual temperature, time-series length and species richness on the rates of thermophilisation. **a**, **e** are for all the taxonomic groups for the terrestrial and the freshwater realms, respectively; (**b**–**d**) and (**f**–**h**) are each taxonomic group: plants, terrestrial insects, and birds (**b**–**d**), and zooplankton, aquatic insects, and fish (**f**–**h**). We also estimated the interaction effects of body size, thermal niche breadth, baseline temperature and temperature change. Effect sizes with gray circles are not significantly different from zero based upon error bars represent 95% confidence intervals are overlapping zero. Please note that the x-axis range differs among the panels. We applied mixed-effects modelling approach (two-sided). For each realm, effect sizes were calculated after removing outliers that lie beyond two standard deviations from the mean, and after accounting for the effects of taxonomic group (random factor), study ID (random factor) and spatial autocorrelation. Sample size information is given in Supplementary Tables S1–S3. Silhouettes were created with BioRender.com.

detected in the terrestrial realm, however, may only delay[13,26], and not ultimately prevent, turnover if temperature change continues. The negative influence of temperature change on terrestrial communities with warmer baseline temperatures highlights that warm sites may already be too warm to experience further thermophilisation, i.e., species adapted to further warming are not available to enter these sites[5,13]. While the data analysed here are extensive, tropical regions (with the highest baseline temperatures) are not well represented as they make up only 0.66% of all time-series due to the paucity of long time-series data, but see[11,40] for communities from tropical regions. These regions harbour highly diverse communities where species likely face greater threats from temperature change due to lower thermal safety margins and smaller average thermal niche breadths[25,41,42]. As biodiversity declines, communities will be more likely to experience thermophilisation, and as a result, temperature change and biodiversity loss may synergistically accelerate rates of community restructuring. Our results are robust to the exclusion of the few tropical communities for which time-series were available, as well the exclusion of phytoplankton and mammal communities, with few exceptions (Supplementary Figs. S4, S5). Nevertheless, we could expect that tropical communities are experiencing faster rates of thermophilisation than temperate communities on average, how thermophilisation varies between freshwater and terrestrial realms in the tropics remain an open question.

## Predictors of thermophilisation at taxonomic level

Different predictors were important in explaining variation in thermophilisation rates among taxonomic groups. Temperature change affected rates of thermophilisation of most taxonomic groups as a main or interactive effect (Fig. 2, Supplementary Table S3). Consistent with previous studies[6,8,9], bird, plant and fish communities that experienced faster temperature change experienced faster thermophilisation rates (Fig. 2b, d, h). Thermal niche breadth modulated the effect of temperature change, resulting in slower thermophilisation rates in terrestrial insect and bird communities, while the interactive effect of temperature change and body size led to faster thermophilisation rates (Fig. 2c, d). This positive interaction with body size is in line with the heat dissipation limit theory[20], which predicts a stronger influence of temperature change on larger bodied species[20] and points towards the metabolic challenges and the higher energy and water requirements of large organisms on land[20–22]. However, fish communities present an interesting case as the separate impacts of temperature change and body size exhibit positive effects on thermophilisation rates, aligning with the predictions of the heat dissipation limit theory[20], but showed a negative interactive influence of temperature change and body size (Fig. 2h). This suggests that the effect of temperature change on the rates of thermophilisation is not constant across different body sizes and that the positive effects of higher temperature change for communities with large body size became negative. By contrast, body size was positively related to thermophilisation rates for plants, birds, fish and aquatic insects, consistent with previous reports where body size was identified as a crucial factor in shaping community responses to temperature change[43] (Fig. 2b, d, g, h). The interaction between body size and thermal niche breadth on thermophilisation rates had mixed effects, being positive for birds and plants (Fig. 2b, d), but negative for aquatic insects and fish (Fig. 2g, h). Additionally, temperature change had a negative impact on the rates of thermophilisation for birds and fish communities with warmer baseline temperatures (Fig. 2d, h), also in line with previous studies[13,44]. For terrestrial insects, species richness tended to reduce the effect of temperature change on thermophilisation (Fig. 2c), again supporting the hypothesis that more diverse communities may be better able to accommodate external changes[28,29]. Lastly, negative rates of thermophilisation were detected in the longer time-series in bird and fish communities (Fig. 2d, h, for overall results for phytoplankton and mammals see Supplementary Table S3).

## Species immigration and extirpation

The effect of species thermal affinities on immigration and extirpation varied across realms. For terrestrial communities, thermophilisation was driven both by the local immigration of species with high thermal affinities relative to extirpated species, and by the local extirpation of species with low thermal affinities compared to persisting species (Fig. 3a). By contrast, extirpated species in the freshwater realm tended to have higher thermal affinities than the persisting taxa (Fig. 3e). The influence of immigrated species with higher thermal affinities than the persisting (Fig. 3b, h, g) or lost (Fig. 3b, c, d, g, h) species was also detected for three and five out of the six taxonomic groups, respectively. Again, planktonic communities were the only exception, in which the extirpated species tended to have higher thermal affinities than the immigrating species (Fig. 3f, Supplementary Fig. S6).

In summary, we found strong and consistent signals of thermophilisation for 6921 communities (excluding phytoplankton and mammals) across realms and multiple habitats, while revealing important differences among taxonomic groups. Planktonic communities were important exceptions among the taxa in our dataset, indicating very strong trends towards the prevalence of cold-affinity species due to loss of warm tolerant species. We currently lack a good understanding of why the composition of planktonic communities is changing towards higher prevalence of cold-affinity species while thermophilisation generally predominates in all other communities. The simplest explanation is that temperature change may have benefitted cold-affinity plankton species that were previously living in environments below their thermal optima (Supplementary Fig. S7). It has been reported that marine phytoplankton typically occur in environments well below their thermal optima[45]. Warming may therefore generally first lead to increased abundances, and probabilities of occurrence and detection, of cold-adapted taxa before it causes them to be lost (temperatures exceeding thermal optima) – leading to thermophilisation. Additionally, with warming, lakes are experiencing the earlier onset and a prolonged period of thermal stratification, which can increase the abundances and overall success of spring bloom, or cold-adapted, taxa[46]. It has also been observed that while the surface water layers of lakes are warming, some lakes in the temperate zone of the northern hemisphere are experiencing cooling in deeper layers[47] which may then have benefited cold-adapted species occupying deeper layers, e.g., the metalimnion species[47]. Finally, it is possible that species that are cold-adapted are also well-adapted to other gradients of environmental change (e.g., nutrient availability), and that the observed pattern of cryophilisation reflects a response to another variable. However, we are currently unable to test these hypothesis with the data at hand.

The analysis of thermophilisation across realms and taxa and interpretation of its possible causes has several challenges. First, we focus on changes in species composition, using occurrence data to estimate CTI rather than abundance data (this decision allowed the inclusion of more community time-series in our analyses). However, CTI estimates based on species abundance may be more sensitive to short-term or seasonal changes in weather[48]. CTI values based on occurrence vs abundance where both data were available were highly correlated (Supplementary Fig. S8). Second, estimation of thermophilisation based on distributional data instead of physiological measurements may have influenced our results. While more detailed physiological data may improve predictions of community turnover, our estimates of thermal preferences were highly correlated with those from other independent estimates of thermal preference[49] (Supplementary Figs. S9–10). Third, for organisms with short generations times, e.g., phytoplankton species, where dynamics and even evolutionary adaptation can be relatively rapid, a greater prevalence of highly resolved community time-series, and more local STI estimates would enable greater confidence in the accuracy of the signal of thermophilisation. Finally, our conclusions may not be

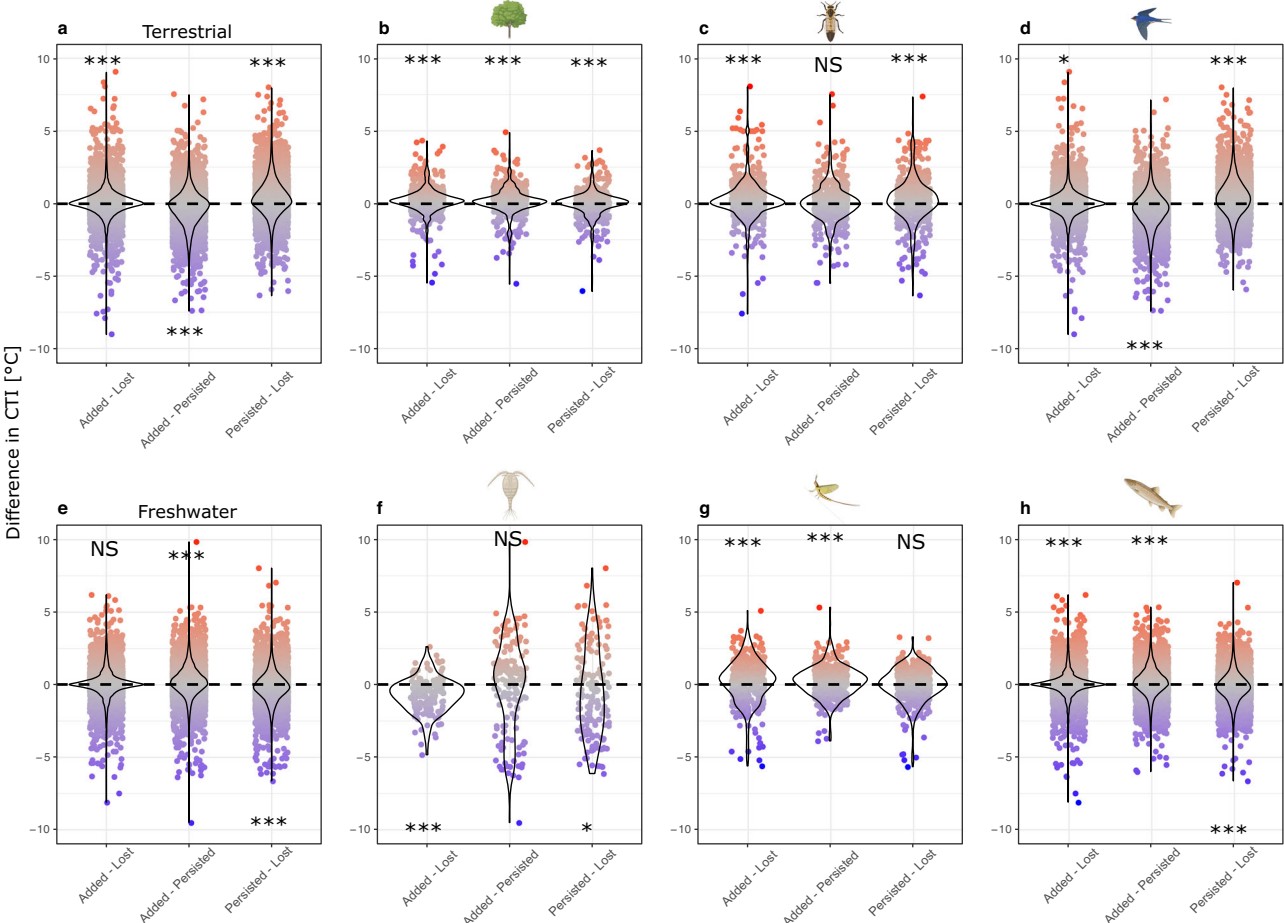

**Fig. 3 | Contributions of immigration and extirpation to thermophilisation.** Difference in the mean thermal affinities of species that immigrated (added), that persisted or that were extirpated (lost) from individual communities. The individual points represent pairwise differences in the mean thermal affinities between these groups (i.e., added, persisted or lost species) for each community. **a**, **e** represent overall results for the terrestrial and the freshwater realms, respectively. All other panels (i.e., b–d, f–h) show the results for each of the six taxonomic groups, indicated by silhouette of each group. Asterisks indicate whether differences are significantly more frequently above or below zero than expected based on a 0.5 probability (binomial-test, one-sided); this is indicated by the position of the asterisks above or below zero on the y-axis. * indicates $P < 0.05$, ** indicates $P < 0.005$, ***indicates $P < 0.001$. Silhouettes were created with BioRender.com.

applicable in tropical ($N = 87$, 0.66%) and high-latitude ($N = 130$; 0.99%) regions as most of the data come from temperate latitudes ($N = 12,843$; 98.6%, Supplementary Fig. S1). Nonetheless, our results were robust to excluding the tropical and polar communities from the data (Supplementary Fig. S4). Lastly, we did not account for the influence of other, often concurrent, anthropogenic environmental changes in our analyses (e.g., eutrophication, re-oligotrophication, land-use change and/or overharvesting), which may have confounding effects beyond temperature change. As an example, previous studies have shown strong influences of land-use change on the rates of thermophilisation for both terrestrial and freshwater communities[9,13,44,50].

Overall, our cross-realm and cross-taxon comparisons revealed thermophilisation occurring in the majority of communities, with the notable exception for lake plankton communities. We detect a strong positive effect of temperature change on thermophilisation rates in both the realms, although the strength of response was weaker in the freshwater realm. Thermophilisation rates were generally positively affected by warming, either as a main effect or via an interaction with other predictors. Importantly, the effects of the biotic predictors of thermophilisation were somewhat idiosyncratic across realms and taxonomic groups. This cautions against assuming homogenous biodiversity responses to climate change. Our findings can be useful to develop strategies to counter the homogenisation of communities.

Various strategies can be employed to address the challenges posed by increasing temperatures in both terrestrial and aquatic systems. For instance, in dense and canopy-covered forests, the impact of rising temperatures on plants and terrestrial species can be mitigated. In aquatic environments, fostering connectivity between water bodies becomes crucial. Minimising the effects of dams and reservoirs within communities is essential to reduce the impacts of elevated temperatures. Furthermore, augmenting riverine and forest cover around water bodies can create microclimatic conditions beneficial for aquatic species, providing a means to cope with rising temperatures. Accounting for these observed differences among realms and taxonomic groups, as well as the impact of local richness, immigration, and extirpation, will be crucial when devising policies to mitigate the influence of climate change on biodiversity.

## Methods
### Compilation of community time-series
We compiled time-series datasets of freshwater and terrestrial communities from several sources: Biodiversity Monitoring Switzerland (BDM, https://biodiversitymonitoring.ch), BioTIME[33], The United States Environmental Protection Agency (EPA)'s National Lakes Assessment (https://nationallakesassessment.epa.gov/), RivFishTIME[34] and several Swiss terrestrial insect studies[35–37]. We filtered the communities on the basis of the following criteria: (1) each community

must include at least four species, (2) the length of the time-series (i.e., end year minus start year) should be of five years or more, (3) thermal affinity information is available for at least 75% species in the community (see section "Thermal affinity calculation" below). Communities differed in the number of species with thermal preference information, ranging from 4 to 399, with only one community containing information for all 666 species. The greatest number of species were found in plant communities in the BDM dataset. We excluded amphibians, reptiles, and freshwater plants as there were fewer than 20 communities in the dataset for these groups. We further excluded communities sampled before 1980, as we were interested in quantifying responses to recent warming and temperature remained relatively stable between 1940 and 1980; this also allowed us to reduce the heterogeneity among time-series in terms of length. We used the unique geographic coordinates of each sampled location to define a community. We avoided aggregating or splitting communities to avoid redefinition and to retain the meaning of community as originally reported. However, to reduce the influence of sampling bias for Bio-TIME and RivFishTIME data, we standardised the number of samples across each time-series by first estimating the minimum samples for each time-series within a year and then standardising sample number to match this minimum number for each time-series following[31,51]. We repeated the resampling process 99 times and took the median values for all variables at each year at each site across these 99 sampled datasets. We also checked and matched the sampling time across time-series where information was available. After filtering the data, we were left with 13,324 communities for our analyses (out of 37,718 communities from our initial data compilation). The final data set included 7123 freshwater and 6201 terrestrial communities and 17,431 species between 1980 and 2019, with an average duration of 13 years for the time-series, with minimum of 5 years and maximum of 38 years.

### Calculation of species temperature affinities (STIs)

We downloaded global distribution data for as many species as possible in our communities from the Global Biodiversity Information Facility (GBIF, https://www.gbif.org/). We downloaded all GBIF occurrence records for each individual taxon and excluded all duplicate and preserved records. We further checked for erroneous coordinates first using CoordinatCleaner[52] R-package and removed the erroneous coordinates. To estimate each species' temperature index (STI, i.e., species thermal affinity), we calculated the mean temperature observed across a species' geographic occurrence range. To calculate the STI and thermal niche breadth for terrestrial species, we used the Chelsa annual mean temperature (Bio01), mean temperature of warmest quarter (Bio10), and mean temperature of coldest quarter (Bio11) layers averaged over 1980 to 2010[53], at a spatial resolution of 1 km$^2$. For aquatic taxa we obtained monthly mean water temperature data from references[54,55] over 1980 to 2010, at a spatial resolution of 10 km$^2$. In summary, the datasets used in this study are based upon one-dimensional water energy routing model (DynWat). This model effectively addresses both energy and water equilibrium on a daily time scale and also takes into account both anthropogenic and the natural processes, such as water dam construction and flooding. It is possible that species may already have shifted their geographic ranges in response to recent climate warming, which could have had an effect on our measures of STI. To check this, we separated the occurrence records in four-time bins i.e., occurrence records before year 1990, between 1991 and 2000, between 2001–2010 and beyond 2010 and found the STI values to be highly correlated (Supplementary Fig. S11). For phytoplankton in particular and in fewer cases for zooplankton, taxonomic identification to species level is not always possible based on morphological characteristics for all species, and in the datasets several taxa were reported only to the genus level. For these taxa we estimated genus-level STIs following reference[2]. For aquatic insects from Swiss BDM data, 19 out of 237 species were reported as belonging

to a species complex in which species cannot be separated based on morphological characteristics. Each complex could contain anywhere from 2 to 33 species. For each complex, we took the mean STI of all species present in the complex following reference[2]. We related STI estimates based upon GBIF with other measures of STI[49,56,57] (Supplementary Figs. S9–S10).

### Community temperature index calculations and estimates of thermophilisation

We calculated the community temperature index (CTI) as the mean of the species temperature indices (STI) for all of the species in the community,

$$CTI = \frac{\sum\limits_{i}^{N} STIi}{N}$$

where N is the number of species in the community. We calculated CTI based on species occurrence rather than weighting by relative abundance, for two main reasons: (1) abundance information was not available for all of our datasets (specifically Swiss BDM data), and (2) it has been previously suggested that the abundance-weighted estimates of CTI are mainly driven by shifts in abundance of the more common species[6,18]. Additionally, abundances are strongly affected by weather conditions of particular years[48]. Hence, analyses based upon occurrence data provide more conservative estimates of community turnover[18]. Additionally, occurrence-based and abundance-weighted estimates are closely related and generate similar rates of thermophilisation (e.g., ref. 9 for freshwater fish, Supplementary Fig. S8). STI is based upon the average environmental temperature experienced by a species across its entire geographic range and is thus a measure of the realised thermal niche of each species. We quantified thermophilisation at each site by taking the slope of a linear regression of CTI as a function of year, where positive slopes indicate shift in communities' composition towards prevalence of warm-affinity species and negative slopes towards more cold-affinity species.

### Limitations on estimates of phytoplankton species temperature indices (STIs) and mammals sample size

Estimates of species thermal affinities based upon geographic occurrence data may be less accurate for phytoplankton than for other taxonomic groups, as suggested by the absences of a strong relationship of mean site-level CTI and mean site-level annual temperatures (Supplementary Fig. S12). This may be because (1) the taxonomic resolution of species identification of phytoplankton is lower on average than for all other taxa – e.g., ~57% of taxa identified only to the genus level and (2) the GBIF records for 676 phytoplankton species out of 1031 had fewer than 5 occurrence records, eliminating more than half of the phytoplankton species from inclusion in the analysis. Though we have no reason to believe that this should systematically bias the analysis towards warm or cold-affinity species, the lack of data should elicit caution when interpreting the phytoplankton results (Supplementary Fig. S13). Additionally, phytoplankton populations have been shown to be able to evolve rapidly in response to local temperature[58,59], and therefore thermal affinities are likely to vary widely across local sites for the same species. As a result, estimates of thermal affinity based on occurrences across a wide geographic range and range of temperatures may not reflect the thermal optima of populations present in any given lake. For such taxa, estimates of thermal affinities for each species based on its local, historical occurrence or abundance records may be preferable. Finally, phytoplankton have short generation times (on the scale of 1–3 days), and most species are microscopic (1–100 μm in size). As a result, they may be responding to temperature variation at much finer temporal and spatial scales than those used to estimate species STIs here. For

Mammals, after rarefaction, we also left with less than 20 time-series. Based upon the points discussed above, we have moved the results for phytoplankton and mammals to the supplementary files.

**Estimates of potential drivers of thermophilisation**
We estimated the rate of temperature change at each site by using the slope of a linear regression of mean annual temperature on year over the time-period over which community change was observed. We downloaded mean monthly temperature data at 1 km$^2$ resolution from the Chelsa dataset[53] and calculated the mean annual temperature for each terrestrial site. For aquatic sites, we downloaded data at 10 km$^2$ resolution from references[54,55] and calculated the mean annual temperature for each site.

We compiled a body size dataset to investigate whether thermophilisation is related to the community average body size. We acquired body size data from several existing data compilations including Fishbase[60], TRY plant database[61], and reference[62] for terrestrial vertebrates. Bird body sizes were most often reported as body mass, so we converted body mass to body size using the data provided in reference[62]. For the remaining taxonomic groups, including insects, phytoplankton and zooplankton, we compiled data directly from the primary literature (list of sources are given in supplement) and from personal data compilations. As measures of body size, we used the longest linear dimension of individuals for phytoplankton and zooplankton, median height for plants, forewing length for dragonflies, moths and aquatic insects, wingspan for butterflies, body length for beetles, the base of tail to the tip of beak for birds, the total length measured from snout to caudal fin margin in the fin lappet for fish, and the tail to snout length for mammals. All body size data was converted to metres. We obtained body size estimates of more than 70% of the taxa identified at the species level in our community data for all taxonomic groups.

We used Bio10 and Bio11 temperature layers from the Bioclim-Chelsa dataset to calculate the thermal niche breadth of each species in the terrestrial realm, by taking the difference between Bio10 and Bio11 for each species. For freshwater species, we used mean monthly temperatures to calculate the thermal niche breadth of each species we took the difference between mean values of the warmest and the coldest quarter.

**Statistical analysis**
Prior to the analysis, we z-score standardised the predictor variables. We also checked the correlation among the predictor variables and found the correlation coefficient well below the tolerant level[63] (Supplementary Fig. S14). To test for differences in the rate of thermophilisation between realms, we modelled thermophilisation as a function of realm, and including study ID (i.e.,. each individual time-series) nested in taxonomic group as random effects, while accounting for spatial autocorrelation using mixed effect models with the "lme" function in the nlme[64] package in R[65]. We added study ID to account for methodological differences that may arise by collating data stemming from different studies or sources. The spatial autocorrelation was included in the model as having a Gaussian variance-covariance structure. We further tested for a different effect of temperature change on the rates of thermophilisation between realms by including an interaction term between temperature change and realm (fixed effects). To test for differences in the rates of thermophilisation among taxonomic groups, we modelled thermophilisation as a function of taxonomic group and included study ID as random effect, investigating each realm separately.

To quantify the influence of our hypothesised predictors on thermophilisation rates at the realm level, we fitted a linear mixed effects model with thermophilisation as a function of temperature change, community mean body size, community mean thermal niche breadth, baseline temperature, time-series length and species richness, and adding interaction terms for all of the fixed effects except time-series length with temperature change. Taxonomic group and study ID were included as random effects. We then fitted mixed effect models for each taxonomic group separately with the same structure and predictors, while accounting for study ID and spatial autocorrelation by adding spatial random effects that follow a Gaussian variance-covariance structure. We additionally added interaction terms between body size and thermal niche breadth because we expected interactions between these drivers[27]. To meet the assumption of the normality of residuals, we excluded all data points that were beyond two standard deviations from the mean of all the residuals of each model[66]. Lastly, we identified the fate of each species in each community as "persisted", "added" or "extirpated" at every time transition. We then tested whether added species tended to have higher mean thermal affinities than species that were extirpated, and similarly, whether species that were extirpated tended to have lower mean thermal affinities than species that persisted at each time transition. To do this, we calculated the mean STI of each group separately (i.e., added, persisted or lost) and compared the groups to one another at each time transition. For example, in added-persisted comparison, we subtracted the average STI of persisted species from the average STI of added species. At the end we took the mean of each group separately across the whole time-series. All analyses were performed in R 4.1.2[65].

**Reporting summary**
Further information on research design is available in the Nature Portfolio Reporting Summary linked to this article.

## Data availability
Part of the data that support the findings of this study are available from Biodiversity Monitoring Switzerland (BDM) but restrictions apply to the availability of these data, which were used under license for the current study, and so are not publicly available. BioTIME, RivFishTime and EPA lakes data are publicly available. The rest of the data is in Figshare: https://figshare.com/s/0d8eaa90cf911a2dcf25. Species level body size data are available from the authors upon request.

## Code availability
The R-code is available on Figshare: https://figshare.com/s/0d8eaa90cf911a2dcf25.

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

## Acknowledgements

We thank the ETH Board for funding through the Blue-Green Biodiversity (BGB) Initiative (BGB2020). This work was also supported by an SNF project grant to AN (310030_197812). LA was supported by the Academy of Finland (grant 340280). FZ was supported by the Swiss National Science Foundation (Grant Number 193645). IRM was supported by an SNSF Postdoc Mobility Fellowship (No. 206844). SRS was supported by the National Science Foundation awards 1915913 and 2227298. We thank the Swiss Federal Office for the Environment (FOEN) for providing plant, butterfly, aquatic insects, and bird occurrence data from the Biodiversity Monitoring Switzerland (BDM). We thank Malin Pinsky and Benjamin Kraemer for providing feedback on the earlier draught. We thank Dirk Karger for assisting us in accessing the Chelsa climate data. We thank Rosi Siber, Luiz Jardim De Queiroz, Sereina Gut, Stefan Pinkert, Dirk Zeuss and Stuart Dennis for helping in compilation of the data. We also thank Edward Lavender for helping in coding. IK was supported by BGB-initiative 2020 as well as by Academic Transition Grant provided by the Eawag, Switzerland.

## Author contributions

A.N., C.R. with the support of F.Z., C.G., M.M.G., I.R.M., J.B., F.P., O.S., N.Z. and B.M. conceived the idea and secured the funding. I.K., A.N., I.R.M., S.G., C.R., T.R., T.S., F.Z., M.R., M.M.G., F.P. and B.M. compiled the data. I.K. conducted the analyses with advice from A.N., C.R., B.M., F.Z., C.G., L.A., S.R.S., F.P., M.M.G, I.R.M., O.S., N.Z., and T. R.. I.K., A.N. and B.M. led the writing with substantial contributions from all co-authors.

## Competing interests

The authors declare no competing interests.

## Additional information

¹Department of Aquatic Ecology, Eawag (Swiss Federal Institute of Aquatic Science and Technology) Überlandstrasse 133, 8600 Dübendorf, Switzerland. ²Swiss Federal Institute for Forest, Snow and Landscape Research (WSL), Flüelastrasse 11, 7260 Davos Dorf, Switzerland. ³Climate Change, Extremes and Natural Hazards in Alpine Regions Research Centre CERC, Flüelastrasse 11, 7260 Davos Dorf, Switzerland. ⁴Department of Zoology, Government (defunct) post-graduate college, Dera Ghazi Khan 32200, Pakistan. ⁵Swiss Federal Institute for Forest, Snow and Landscape Research (WSL), Zürcherstrasse 111, 8903 Birmensdorf, Switzerland. ⁶ETH Zurich, Department of Environmental Systems Science, Institute of Terrestrial Ecosystems, 8092 Zurich, Switzerland. ⁷Institute for Biodiversity and Ecosystem Dynamics, University of Amsterdam, 1090 GE Amsterdam, The Netherlands. ⁸University of London, Queen Mary, London, UK. ⁹Research Centre for Ecological Change, Organismal and Evolutionary Biology Research Programme, University of Helsinki, PO Box 65 (Viikinkaari 1), 00014 Helsinki, Finland. ¹⁰Department of Fish Ecology and Evolution, Eawag (Swiss Federal Institute of Aquatic Science and Technology), Seestrasse 79, 6047 Kastanienbaum, Switzerland. ¹¹Department of Environmental Sciences, Zoology, University of Basel, Vesalgasse 1, 4051 Basel, Switzerland. ¹²Department of Evolutionary Biology and Environmental Studies, University of Zurich, Zurich, Switzerland. ¹³Division of Aquatic Ecology and Evolution, Institute of Ecology and Evolution, University of Bern, Baltzerstrasse 6, 3012 Bern, Switzerland. ¹⁴Hintermann & Weber AG Austrasse 2a, 4153 Reinach, Switzerland. ¹⁵Swiss Ornithological Institute, Seerose 1, 6204 Sempach, Switzerland. ¹⁶Denison University, Data Analytics Program, Granville, OH 43023, USA. ¹⁷Conservation Genetics Group, Senckenberg Research Institute and Natural History Museum Frankfurt, 63571 Gelnhausen, Germany. ¹⁸Faculty of Biological Sciences, Institute for Ecology, Evolution and Diversity, Goethe University, Max-von-Laue-Straße 9, 60438 Frankfurt am Main, Germany. ✉e-mail: imrankhaliq9@hotmail.com; anita.narwani@eawag.ch

