## [Peer Review File · Nature Communications]

Warming underpins community turnover in temperate freshwater and terrestrial communitiesEditorial Note: This manuscript has been previously reviewed at another journal that is not operating a transparent peer review scheme. This document only contains reviewer comments and rebuttal letters for versions considered at *Nature Communications*.

Reviewer #5 (Remarks to the Author):

This is a novel and original study of broad interest. It uses an impressive dataset of community time series to compare the rates of thermophilisation among different taxonomic groups from freshwater and terrestrial realms. While there were no differences between the realms or taxonomic groups, the authors found different independent and interactive impacts of a suite of predictors on the rate of thermophilisation. Among the key findings, the analysis suggested that freshwater communities dominated by large species with wider thermal niches might be more tolerant to warming than their terrestrial counterparts, especially when the temperature changes are relatively small. The prevalence of cold-affinity and loss of warm tolerant species with warming in planktonic communities is puzzling and intriguing. Yet, the proposed mechanisms by the authors (Line 233-253) are plausible and very interesting avenue for future research. The methodology is sound and the statistical analyses are rigorous and robust. The manuscript already underwent a review process and improved; it is very well written and easy to follow, even though the use of subheadings would improve/highlight the study structure (currently very long, continuous text). I only have relatively minor comments are suggestions, which I hope can further strengthen the manuscript.

Although it is briefly mentioned in the introduction, it should be elaborated in more detail what are the differences between freshwater and terrestrial realms and why would we expect realm differences in thermophilisation in the first place. In my opinion, one of the key differences are the plastic responses in ectotherm body size to warming. Aquatic organisms often mature at smaller body size, but the effect is much weaker or absent for terrestrial ectotherms in warmer environments – see for instance Horne et al. (2015): <https://doi.org/10.1111/ele.12413>.

The sentence “Our results are robust to the exclusion of tropical communities...” (line 188) sounds vague and I would suggest explaining more explicitly what you mean here. I would also explain whether missing tropical data lead to overestimates or underestimates of your thermophilisation rates, whether this might differ between the contrasted realms and why.

In the introduction, the authors suggest that the systematic analysis of the drivers of thermophilisation (as performed in this study) could lead to the development of more effective conservation and management strategies. In the final paragraph, can you please elaborate on which conservation and management strategies would you recommend based on your findings?

Stability is a complex multifaceted concept, and if used here (Line 102), it should be clearly defined what kind of stability is considered.

The authors explain that CTI is calculated as the mean of STI's, “based on species’ presence in the community” (Line 123). This calculation does not account for different densities and the presence of a dominant species is equivalent to a rare singleton of another species. However, in the Discussion, the authors discuss well the advantages and disadvantages of their approach (in general) and (more specifically) they found the results to be highly correlated for analyses based on occurrence vs abundance data.

Line 83: Please delete “by”

Line 133: Change “indicate” to “indicating” or “which indicates”

Lines 169-172: It looks like the effect of temperature change on thermophilisation is also mitigated by

thermal niche breath (Fig. 2a), which was missed in this description here.

Lines 202-205: Cumbersome sentence – can this be streamlined or rephrased?

Lines 218-220: This sentence needs to be rephrased because it is not the “time-series”, which has this impact.

Line 226: Change “incoming” to “immigrated”

Line 253: Change to “test these hypotheses”

Line 274: Change “compounding” to “confounding”

Figure 3 – Please add y-axis labels

Reviewer #6 (Remarks to the Author):

Review of response to reviewer #3

As requested by the Editor, I focus my review on how the authors have responded to reviewer 3 who is not available. I have not reviewed the paper previously.

In my opinion, the authors have adequately responded to all the comments.

Main outstanding comments:

(1) Reviewer 3 comment: “Nonetheless, I was not really convinced by the authors’ responses and supplementary analyses (and figures) to my initial comment on using data from the period 1980-2010, which already incorporate climate warming, to compute STI values” and “If the correlation values among those three non-overlapping periods are still close to 0.9 for both terrestrial and freshwater species, then I would be convinced that this potential methodological thermal niche shifts is minor and not playing any role in the main results.”

Assessment of author response: They have compared how the STI values would vary using different subsets of data and show that the STIs are highly correlated (>0.95) across different data period choices. Based on this, Reviewer 3 should be satisfied that their data period choice is unimportant.

(2) Reviewer 3 comment: “Additionally, another point I raised in my original assessment and that was not addressed by the authors is about the relevance of anthropogenic disturbances, such as land-use changes.” And “I am not saying that anthropogenic pressures, like land-use changes, should be accounted for explicitly in the authors’ models, but I expect that the authors will somehow discuss their main findings in light of alternative drivers such as more direct anthropogenic disturbances.”

Assessment of author response: The authors have added several sentences in the Discussion on how land-use impacts were not considered and that this can affect the rates of thermophilization. This seems to meet the reviewers request.

(3) Reviewer 3 comment: “Finally, given the clear geographical bias in the dataset used by the authors (cf. maps in Extended Data Fig. 1), I think the authors should tone down a bit their findings by reminding the reader that these results are mostly true for temperate systems only”.

Assessment of author response: The authors have added temperate to the title and qualified this at various points in the text. Their response is satisfactory.

Reviewers' specific comments:

Line 1 comment: Authors have followed the suggestion of the reviewer.

Line 80 comment: Authors have followed the suggestion of the reviewer.

Line 97-100 comment: The effect of body size could be argued either way. I think it is appropriate for the authors to argue one way without an exhaustive list of all possible mechanisms leading to other relationships.

Lines 103-100 comment: The author clarify that they test the interaction between body size and thermal niche breath.

Line 110-110 comment: The author clarify that warmer base-line temperatures might be associated with slower rates of thermophilization.

Line 155-156 comment: Authors have followed the suggestion of the reviewer.

Line 168 comment: Authors have followed the suggestion of the reviewer.

Line 176 comment: Original sentence has been deleted.

Line 206 comment: Authors have followed the suggestion of the reviewer.

Line 237 comment: Authors have followed the suggestion of the reviewer.

Line 439-440 comment: It seems reasonable for the authors to focus on change since 1980 since there are fewer data before this period, and less change in temperatures.

Line 455 comment: Authors have followed the suggestion of the reviewer.

Line 462-463 comment: Authors have clarified the query.

Line 471 comment: Authors have clarified the query.

Line 473-475: Phytoplankton results have been moved to the SI to avoid impacts of taxonomic bias.

Line 578-582 comment: Authors have followed the suggestion of the reviewer.

REVIEWERS' COMMENTS

Reviewer #5 (Remarks to the Author):

This is a novel and original study of broad interest. It uses an impressive dataset of community time series to compare the rates of thermophilisation among different taxonomic groups from freshwater and terrestrial realms. While there were no differences between the realms or taxonomic groups, the authors found different independent and interactive impacts of a suite of predictors on the rate of thermophilisation. Among the key findings, the analysis suggested that freshwater communities dominated by large species with wider thermal niches might be more tolerant to warming than their terrestrial counterparts, especially when the temperature changes are relatively small. The prevalence of cold-affinity and loss of warm tolerant species with warming in planktonic communities is puzzling and intriguing. Yet, the proposed mechanisms by the authors (Line 233-253) are plausible and very interesting avenue for future research. The methodology is sounds and the statistical analyses are rigorous and robust. The manuscript already underwent a review process and improved; it is very well written and easy to follow, even though the use of subheadings would improve/highlight the study structure (currently very long, continuous text). I only have relatively minor comments are suggestions, which I hope can further strengthen the manuscript.

REPLY: *Now, we have added the subheadings.*

Although it is briefly mentioned in the introduction, it should be elaborated in more detail what are the differences between freshwater and terrestrial realms and why would we expect realm differences in thermophilisation in the first place. In my opinion, one of the key differences are the plastic responses in ectotherm body size to warming. Aquatic organisms often mature at smaller body size, but the effect is much weaker or absent for terrestrial ectotherms in warmer environments – see for instance Horne et al.

(2015): <https://doi.org/10.1111/ele.12413>.

REPLY: *We have added more sentences about differences in context of body size and now it reads as on lines # 92-100: "First, large species will have stronger responses to rising temperatures compared to small species due to their greater metabolic constraints, and higher water and energy requirements for thermoregulation²⁰⁻²². Moreover, the correlation between body size and temperature exhibits a more pronounced negative trend in aquatic ectotherms compared to their terrestrial counterparts, indicating notable distinctions between these two ecological realms²³. Consequently, we anticipate a stronger positive association between mean body size of species in aquatic communities and temperature compared to the relationship terrestrial communities, resulting in faster rates of thermophilisation in the former.²⁴"*

The sentence "Our results are robust to the exclusion of tropical communities..." (line 188) sounds vague and I would suggest explaining more explicitly what you mean here. I would also explain whether missing tropical data lead to overestimates or underestimates of your thermophilisation rates, whether this might differ between the contrasted realms and why.

REPLY: We have added this and lines # 194-199 read as: "Our results are robust to the exclusion of the few tropical communities for which time-series were available, as well the exclusion of phytoplankton and mammal communities, with few exceptions (Extended Data Figs. 4, 5). Nevertheless, we could expect that tropical communities are experiencing faster rates of thermophilisation than temperate communities on average, due to narrower thermal safety margins. Due to the paucity of tropical time-series, how thermophilisation varies between freshwater and terrestrial realms in the tropics remains an open question."

In the introduction, the authors suggest that the systematic analysis of the drivers of thermophilisation (as performed in this study) could lead to the development of more effective conservation and management strategies. In the final paragraph, can you please elaborate on which conservation and management strategies would you recommend based on your findings?

REPLY: We have added this in conclusion section and it read as: ". Our findings can be useful to develop strategies to counter the homogenisation of communities. Various strategies can be employed to address the challenges posed by increasing temperatures in both terrestrial and aquatic systems. For instance, in dense and canopy-covered forests, the impact of rising temperatures on plants and terrestrial species can be mitigated. In aquatic environments, fostering connectivity between water bodies becomes crucial. Minimizing the effects of dams and reservoirs within communities is essential to reduce the impacts of elevated temperatures. Furthermore, augmenting riverine and forest cover around water bodies can create microclimatic conditions beneficial for aquatic species, providing a means to cope with rising temperatures. Additionally, accounting for these observed differences among realms and taxonomic groups, as well as the impact of local richness, immigration, and extirpation, will be crucial when devising policies to mitigate the influence of climate change on biodiversity."

Stability is a complex multifaceted concept, and if used here (Line 102), it should be clearly defined what kind of stability is considered.

REPLY, Line # 105: We now have changed the stability to resistance as we meant that communities will be resistant to change.

The authors explain that CTI is calculated as the mean of STI's, "based on species' presence in the community" (Line 123). This calculation does not account for different densities and the presence of a dominant species is equivalent to a rare singleton of another species. However, in the Discussion, the authors discuss well the advantages and disadvantages of their approach (in general) and (more specifically) they found the results to be highly correlated for analyses based on occurrence vs abundance data.

Line 83: Please delete "by"

REPLY: Done

Line 133: Change "indicate" to "indicating" or "which indicates"

REPLY: Line # 142, change to "indicating".

Lines 169-172: It looks like the effect of temperature change on thermophilisation is also mitigated by thermal niche breadth (Fig. 2a), which was missed in this description here.

REPLY: *We now have added this and it now reads as: "Furthermore, on land the impact of temperature change on thermophilisation rates was influenced by baseline temperature, thermal niche breadth and species richness, with warmer baseline conditions, broader thermal niche breadth and greater species richness mitigating the effect of temperature change (i.e. underpinning slower rates of thermophilisation; Fig. 2a)"*

Lines 202-205: Cumbersome sentence – can this be streamlined or rephrased?

REPLY, lines # 213-216: *We have modified this sentence and it reads as: "However, fish communities present an interesting case as the separate impacts of temperature change and body size exhibit positive effects on thermophilisation rates, aligning with the predictions of the heat dissipation limit theory²⁰, but showed a negative interactive influence of temperature change and body size (Fig. 2h)."*

Lines 218-220: This sentence needs to be rephrased because it is not the "time-series", which has this impact.

REPLY: *We agree and now have modified this sentence and it reads as on lines # 229-232: ". Lastly, negative rates of thermophilisation were detected in the longer time-series in bird and fish communities (Fig. 2d, h, for overall results for phytoplankton and mammals see Extended Data Table 3)."*

Line 226: Change "incoming" to "immigrated"

REPLY: *Done, line # 238*

Line 253: Change to "test these hypotheses"

REPLY: *Done, line # 265*

Line 274: Change "compounding" to "confounding"

REPLY: *Done, line # 286*

Figure 3 – Please add y-axis labels

REPLY: *Added*

Reviewer #6 (Remarks to the Author):

Review of response to reviewer #3

As requested by the Editor, I focus my review on how the authors have responded to reviewer 3 who is not available. I have not reviewed the paper previously.

In my opinion, the authors have adequately responded to all the comments.

Main outstanding comments:

(1) Reviewer 3 comment: "Nonetheless, I was not really convinced by the authors' responses and supplementary analyses (and figures) to my initial comment on using data from the period 1980-2010, which already incorporate climate warming, to compute STI values" and "If the correlation values among those three non-overlapping periods are still close to 0.9 for both terrestrial and freshwater species, then I would be convinced that this potential methodological thermal niche shifts is minor and not playing any role in the main results."

Assessment of author response: They have compared how the STI values would vary using different subsets of data and show that the STIs are highly correlated (>0.95) across different data period choices. Based on this, Reviewer 3 should be satisfied that their data period choice is unimportant.

REPLY: *Thank you!*

(2) Reviewer 3 comment: "Additionally, another point I raised in my original assessment and that was not addressed by the authors is about the relevance of anthropogenic disturbances, such as land-use changes." And "I am not saying that anthropogenic pressures, like land-use changes, should be accounted for explicitly in the authors' models, but I expect that the authors will somehow discuss their main findings in light of alternative drivers such as more direct anthropogenic disturbances."

Assessment of author response: The authors have added several sentences in the Discussion on how land-use impacts were not considered and that this can affect the rates of thermophilization. This seems to meet the reviewers request.

REPLY: *Thank you!*

(3) Reviewer 3 comment: "Finally, given the clear geographical bias in the dataset used by the authors (cf. maps in Extended Data Fig. 1), I think the authors should tone down a bit their findings by reminding the reader that these results are mostly true for temperate systems only".

Assessment of author response: The authors have added temperate to the title and qualified this at various points in the text. Their response is satisfactory.

REPLY: *Thank you!*

Reviewers' specific comments:

Line 1 comment: Authors have followed the suggestion of the reviewer.

REPLY: *Thank you!*

Line 80 comment: Authors have followed the suggestion of the reviewer.

REPLY: *Thank you!*

Line 97-100 comment: The effect of body size could be argued either way. I think it is

appropriate for the authors to argue one way without an exhaustive list of all possible mechanisms leading to other relationships.

REPLY: *Following other reviewer`s suggestions, this part has been modified and now we have highlighted the body size differences in the two realms. The whole section reads now as: "First, large species will have stronger responses to rising temperatures compared to small species due to their greater metabolic constraints, and higher water and energy requirements for thermoregulation²⁰⁻²². Moreover, the correlation between body size and temperature exhibits a more pronounced negative trend in aquatic ectotherms compared to their terrestrial counterparts, indicating notable distinctions between these two ecological realms²³. Consequently, we anticipate a more robust positive association in the mean body size of species within aquatic communities as opposed to terrestrial communities, resulting in faster rates of thermophilisation in the former²⁴"*

Lines 103-100 comment: The author clarify that they test the interaction between body size and thermal niche breath.

REPLY: *Thank you!*

Line 110-110 comment: The author clarify that warmer base-line temperatures might be associated with slower rates of thermphilization.

REPLY: *Thank you!*

Line 155-156 comment: Authors have followed the suggestion of the reviewer.

REPLY: *Thank you!*

Line 168 comment: Authors have followed the suggestion of the reviewer.

REPLY: *Thank you!*

Line 176 comment: Original sentence has been deleted.

Line 206 comment: Authors have followed the suggestion of the reviewer.

REPLY: *Thank you!*

Line 237 comment: Authors have followed the suggestion of the reviewer.

REPLY: *Thank you!*

Line 439-440 comment: It seems reasonable for the authors to focus on change since 1980 since there are fewer data before this period, and less change in temperatures.

REPLY: *Thank you!*

Line 455 comment: Authors have followed the suggestion of the reviewer.

REPLY: *Thank you!*

Line 462-463 comment: Authors have clarified the query.

REPLY: *Thank you!*

Line 471 comment: Authors have clarified the query.

REPLY: Thank you!

Line 473-475: Phytoplankton results have been moved to the SI to avoid impacts of taxonomic bias.

Line 578-582 comment: Authors have followed the suggestion of the reviewer.

REPLY: Thank you!